# Relationship between Lipid Profiles and Glycemic Control Among Patients with Type 2 Diabetes in Qingdao, China

**DOI:** 10.3390/ijerph17155317

**Published:** 2020-07-23

**Authors:** Shukang Wang, Xiaokang Ji, Zhentang Zhang, Fuzhong Xue

**Affiliations:** 1Department of Biostatistics, School of Public Health, Cheeloo College of Medicine, Shandong University, 44, Wenhuaxi Street, Jinan 250012, Shandong, China; wsk2001@sdu.edu.cn (S.W.); jxk@sdu.edu.cn (X.J.); 2Institute for Medical Dataology, Shandong University, 12550, Erhuandong Street, Jinan 250002, Shandong, China; 3Qingdao West Coast New District Center for Disease Control and Prevention, 567, Lingshanwan Street, Huangdao District, Qingdao 266400, China; jnjkzzt@163.com

**Keywords:** China, cross-sectional study, lipid profiles, glycemic control

## Abstract

Glycosylated hemoglobin (HbA1c) was the best indicator of glycemic control, which did not show the dynamic relationship between glycemic control and lipid profiles. In order to guide the health management of Type 2 diabetes (T2D), we assessed the levels of lipid profiles and fasting plasma glucose (FPG) and displayed the relationship between FPG control and lipid profiles. We conducted a cross-sectional study that included 5822 participants. Descriptive statistics were conducted according to gender and glycemic status respectively. Comparisons for the control of lipid profiles were conducted according to glycemic control. Four logistic regression models were generated to analyze the relationship between lipid profiles and glycemic control according to different confounding factors. The metabolic control percentage of FPG, triglyceride (TG), total cholesterol (TC), high density lipoprotein cholesterol (HDL-C) and low density lipoprotein cholesterol (LDL-C) was 27.50%, 73.10%, 28.10%, 64.20% and 44.80% respectively. In the fourth model with the most confounding factors, the odds ratios (ORs) and 95% confidence intervals (CIs) of TG, TC, LDL-C and HDL-C were 0.989 (0.935, 1.046), 0.862 (0.823, 0.903), 0.987 (0.920, 1.060) and 2.173 (1.761, 2.683). TC and HDL-C were statistically significant, and TG and LDL-C were not statistically significant with adjustment for different confounding factors. In conclusion, FPG was significantly associated with HDL and TC and was not associated with LDL and TG. Our findings suggested that TC and HDL should be focused on in the process of T2D health management.

## 1. Introduction

The primary task of health management in Type 2 diabetes (T2D) patients is to prevent diabetes-related complications. Previous studies have shown that good control of lipid profiles and glycemic levels can effectively prevent complications such as cardiovascular disease, diabetic nephropathy and diabetic retinopathy [1,2]. Lipid profiles referred to lipids in plasma, generally including triglyceride (TG), total cholesterol (TC), high density lipoprotein (HDL), and low density lipoprotein (LDL) clinically [3]. For patients with cardiovascular disease and T2D, lipid profiles should be strictly controlled to reduce mortality and complications [4]. According to the American Diabetes Association guidelines, glycemic control needed to be determined based on levels of self-monitoring of blood glucose (SMBG) and glycosylated hemoglobin (HbA1c). As a gold standard for evaluating long-term glycemic control, HbA1c < 7.0 was clinically defined as glycemic control [5]. Moreover, there is a close relationship between glycemic control and control of lipid profiles. On one hand, good control of lipid profiles was one of the important factors influencing glycemic control in patients with T2D [6]. For example, infusion therapy of HDL increased plasma high density lipoprotein cholesterol (HDL-C) levels and reduced plasma glucose levels in T2D patients by increasing plasma insulin and activating AMP-activated protein kinase in skeletal muscles [7]. On the other hand, good glycemic control contributed to control of lipid profiles for patients with T2D. Fujita Y, et al. reported that short-term intensive glycemic control could significantly decrease levels of TC by improving lipid metabolism [8]. Therefore, good control of glycemic levels and lipid profiles are very important as well as being complicated in patients with T2D.

Although many studies have confirmed the relationship between glycemic control and lipid profiles in patients with T2D, the results are quite inconsistent [9,10,11]. Khan HA et al. reported that the level of HbA1c was positively correlated with TG, TC, and LDL-C, but negatively correlated with HDL-C [10]. However, a cross-sectional study in Eastern Sudan showed that poor glycemic control was not associated with TG but was associated with high TC levels [11]. Furthermore, some studies showed that HbA1c was associated with TG and TC, rather than LDL-C and HDL-C [12].

It is certain that the relationship between glycemic control and lipid profiles in T2D patients is a major problem that needs to be solved urgently. HbA1c was used as an indicator of glycemic control in most previous studies [13,14]. HbA1c was a stable indicator of glycemic control, which reflected an average glycemic level of about three months and did not show glycemic variability over a period of time [15]. Fasting plasma glucose (FPG) is an economical and important indicator for the diagnosis and glycemic variability of T2D. Few studies have shown the relationship between FPG control and lipid profiles. Therefore, this study aims to assess the levels of lipid profiles and FPG, and investigate the relationship between FPG control and lipid profiles in patients with T2D in Qingdao City, Shandong Province, China.

## 2. Materials and Methods

### 2.1. Study Population

The subjects in this study were from the Multicenter Longitudinal Health Management Cohorts in Shandong province, China, which included a total of 15,369 patients from T2D Health Management System of Huangdao District Center for Disease Control and Prevention, Qingdao, China. The T2D health management system was established in 2009 in Huangdao District, Qingdao City. A total of 15,369 T2D patients were included in the management system by July 2015. The number of patients entering the health management system in 2009, 2010, 2011, 2012, 2013, 2014 and 2015 were 1457, 65, 57, 493, 102, 12,318 and 877 respectively. According to the inclusion criteria, patients aged 18 years or older were included, while they must have complete demographic and clinical data. Exclusion criteria were an age younger than 18 years, pregnant women, breast feeding, malignancies, and autoimmune diseases. The 15,369 T2D patients who entered the health management system had met the above inclusion and exclusion criteria. Due to the absence of information related to demographic and clinical variables, a total of 5822 participants was included in the final analysis. This study was approved by the Ethics Committee of the School of Public Health, Shandong University (Code NO.: 20140322).

### 2.2. Investigation and Measurements

During the T2D patient’s entry into the management system, trained interviewers administered a standardized questionnaire to obtain demographic characteristics, including age, gender, current smoking status (yes or no), current alcohol intake (not drinking, occasionally, often), marital status (yes or no), educational level (uneducated, primary or junior high school, high school, and above), and regular exercise frequency (not exercising, 1–3 times per month, 1–2 times per week, ≥3 times per week). After an overnight fast of at least 12 h, all subjects underwent a standardized medical examination that included routine anthropometric, clinical, and laboratory tests. The anthropometric measurements involved height, weight, SBP, and DBP. Two blood pressure values were taken about 10 min apart on the right arm by trained examiners using a mercury sphygmomanometer. Height and weight were measured after the participants had removed their shoes, heavy clothing, and belts. Body mass index (BMI) was calculated as weight (kg)/height^2^ (m). Duration of T2D was calculated as entry time minus diagnosis time of T2D. Laboratory tests included TG, TC, LDL-C, HDL-C, and FPG. Fasting blood samples were taken from the T2D patients and the serum was separated from the blood in a centrifuge tube. The serum samples were tested within two hours or frozen at −20 °C until testing. FPG was measured using the hexokinase method or the glucose oxidase method. The enzymatic method was used to measure TG and TC. HDL-C and LDL-C were measured using the direct test method. Automatic biochemical detectors (Olympus AU400 and Hitachi 7600, Tokyo, Japan) were used to test the concentration of TG, TC, HDL-C, LDL-C, and FPG.

### 2.3. Definitions of Glycemic Control Lipid Profiles Control and Hypertension

FPG < 7.0 mmol/L and FPG ≥ 7.0 mmol/L were defined as good glycemic control and poor glycemic control, respectively.

According to Standards of medical care for T2D in China 2019 [16], the control of each lipid component was defined as follows: TG < 1.70 mmol/L and TG ≥ 1.70 mmol/L were defined as controlled and uncontrolled; TC < 4.50 mmol/L and TC ≥ 4.50 mmol/L were defined as controlled and uncontrolled; HDL-C > 1.00 mmol/L and HDL-C ≤ 1.00 mmol/L were defined as controlled and uncontrolled for male; and HDL-C > 1.30 mmol/L and HDL-C ≤ 1.30 mmol/L for females. LDL-C < 2.6 mmol/L and LDL-C ≥ 2.60 mmol/L were defined as controlled and uncontrolled for T2D patients without atherosclerotic cardiovascular disease, and LDL-C < 1.80 mmol/L and LDL-C ≥ 1.80 mmol/L for T2D patients with atherosclerotic cardiovascular disease.

Hypertension was defined as an SBP ≥ 140 mmHg or a DBP ≥ 90 mmHg or the self-reported current use of antihypertensive medication [17].

### 2.4. Statistical Analysis

Descriptive statistics were conducted according to gender and glycemic control status respectively. The numerical variables were described as the median (*M*) and quartile range (*Q*) based on the skewed distribution. The categorical variables were expressed as number and percentage. Various characteristics were compared between the glycemic control group and uncontrolled group. Mann–Whitney U tests were used to test the difference of the continuous variables and ordinal variables. χ^2^ tests were adopted to test the difference of categorical variables. Comparisons using χ^2^ tests for the control of lipid profiles (TG, TC, HDL-C, and LDL-C) were conducted according to glycemic control. Logistic regression analyses were further performed to estimate the odd ratios (ORs) and 95% confidence intervals (CIs) of good glycemic control. Four logistic regression models were conducted to analyze the relationship between lipid profiles and glycemic control according to the adjustment for different confounding factors. In the four logistic regression models, the outcome variables were whether FPG (FPG < 7.0 = 1, FPG ≥ 7.0 = 0) was controlled. Our main covariates of interest were TG, TC, HDL-C, and LDL-C. The four lipid profile variables entered the logistic model as numerical variables. No confounding factors were adjusted for in the first model; age and gender were adjusted for in the second model. In the third model, the adjusted confounders were extended to include age, gender, smoking, alcohol drinking, marital status, educational level, and physical exercise frequency. Compared with the third model, BMI, hypertension, duration of T2D, and antidiabetics were further added to the fourth model. SPSS 23.0 software (IBM Corp., Armonk, NY, USA) was used to perform all statistical analyses.

## 3. Results

A total of 5822 subjects were included in this study, which included 2188 males and 3634 females. The characteristics of the subjects were summarized in Table 1. The median age of both men and women was 67.00, and the quartile range of age was 16.00 and 14.00 respectively.

Table 2 showed the distributions of age, BMI, SBP, DBP, TG, TC, HDL-C, LDL-C, and duration of T2D and the prevalence of male, smoking, alcohol drinking, hypertension, antidiabetics, in marriage, educational level, and physical exercise frequency according to glycemic control. There was a statistically significant difference in age, SBP, DBP, TC, HDL-C, hypertension, antidiabetics, and educational level.

Table 3 shows that the metabolic control percentage of FPG, TG, TC, HDL-C, and LDL-C for all T2D patients was 27.50%, 73.10%, 28.10%, 64.20%, and 44.80% respectively. There was a statistically significant difference in the prevalence of TC and HDL-C between good glycemic control and poor glycemic control groups.

Table 4 showed the relationship between lipid profiles and glycemic control with adjusting the different confounding factors. In the first model, the odds ratios (ORs) and 95% confidence intervals (CIs) of TG, TC, LDL-C and HDL-C were 0.984 (0.932, 1.039), 0.853 (0.816, 0.891), 0.998 (0.932, 1.068), and 2.229 (1.817, 2.735) respectively, without adjustment for any variables. In the second model, with adjustment for age and gender, the ORs and 95% CIs of TG, TC, LDL-C, and HDL-C were 0.978 (0.926, 1.034), 0.835 (0.798, 0.873), 0.981 (0.916, 1.051), and 2.233 (1.818, 2.743). In the third model, with adjustment for age, gender, smoking, alcohol drinking, marital status, educational level, and physical exercise frequency, the ORs and 95% CIs of TG, TC, LDL-C, and HDL-C were 0.966 (0.913, 1.022), 0.840 (0.803, 0.879), 0.967 (0.902, 1.036), and 2.223 (1.809, 2.733). In the fourth model, with adjustment for age, gender, smoking, alcohol drinking, marital status, educational level, physical exercise frequency, BMI, hypertension, duration of T2D, and antidiabetics, the ORs and 95% CIs of TG, TC, LDL-C, and HDL-C were 0.989 (0.935, 1.046), 0.862 (0.823, 0.903), 0.987 (0.920, 1.060), and 2.173 (1.761, 2.683). In the four models, TC and HDL-C were statistically significant, and TG and LDL-C were not statistically significant.

## 4. Discussion

T2D patients had an increased risk of death and disability due to associated complications, suggesting the significance of good control for lipid profiles and glycemic level [18]. Therefore, the rates of FPG control and lipid profile control were evaluated, and the relationships between FPG control and lipid profiles were analyzed for T2D patients in this study. The results showed that the rates of FPG control and lipid profiles control were low, and that FPG control was significantly associated with HDL and TC.

In this study, the control rates of FPG, TG, TC, HDL-C, and LDL-C for all T2D patients were 27.50%, 73.10%, 28.10%, 64.20%, and 44.80% respectively, with especially lower control rates for FPG and TC in T2D patients. A study from Beijing, China, showed a glycemic control rate of 30% [19]. Furthermore, a study from Xinjiang, China, showed that the late onset T2D patients had a glycemic control rate of 23.1%, while the early onset T2D patients had a glycemic control rate of only 14.2%. Moreover, both patients had lipid control rates of less than 10% [20]. The patients with T2D from nine European countries had a glycemic control rate of only 37.4% [21]. The T2D patients from America had a higher glycemic control rate than other countries, reaching about 50% [22]. The abnormal rates of HbA1c, TG, TC, HDL-C, and LDL-C were 82.6%, 69.5%, 40.7%, 41%, and 54.1% respectively for the T2D patients in Pakistan [23]. All these results showed that poor control of blood glucose and lipid profiles in T2D patients was a problem to be solved [24,25].

Moreover, the results of this study showed that the T2D patients with an FPG value ≥ 7.0 had a significant increase in TC and a significant decrease in HDL-C, rather than TG and LDL-C, compared with the T2D patients with an FPG value < 7.0. Prospective studies confirmed that lower HDL-C was not only associated with future high incidence of T2D [26,27], but is also bad for glycemic control in T2D patients [28]. This study was also consistent with the conclusion that lower HDL-C was not good for glycemic control.

Similar to our findings, Mullugeta Y et al. reported that HbA1c was positively associated with TC while not with LDL [29]. However, some studies showed a positive correlation between HbA1c and LDL and TC [30]. Certainly, many studies have questioned the relationship between HbA1c and LDL. Some studies reported that LDL was not associated with HbA1c [31] and T2D [32,33], and even that lower LDL could increase the incidence of T2D [34,35]. Different from our results, previous studies have shown that elevated TG levels are strongly associated with poor glycemic control in T2D patients [36,37].

Unlike our findings, some studies have shown that T2D patients with an HbA1c value ≥ 7.0% have a significant increase in TC and LDL-C without significant changes in TG and HDL-C compared with the T2D patients with an HbA1c value < 7.0% [38]. A study from Sudan reported there was no significant difference in TG, TC, LDL, and HDL between the glycemic control group and the uncontrolled group [39]. However, a study from Montenegro showed that HbA1c was associated with TG, TC, LDL, and HDL [40]. These inconsistent results may be partly due to the relative stability of HbA1c. That is, HbA1c levels are stable over a period of time, while lipid profiles and FPG levels are dynamically changing. Then, studies on the relationship between HbA1c and lipid profiles at different time points over a period of time may present different results.

Of course, FPG and lipid profiles levels of T2D patients are affected by many factors. The relationship between glycemic control and lipid profiles may be related to lipid-lowering agents used in T2D patients [41]. Some lipid-lowering medications can improve blood glucose metabolism [42]. Statins were associated with an increase in HbA1c compared with placebo in T2D patients [43]. Metformin monotherapy appreciably improved dyslipidemia in statin-naive patients with T2D [44]. Agents for T2D have an effect on lipid profiles and conversely, agents controlling lipid profiles have an effect on FPG. T2D comorbidities including cardiovascular diseases (CVD) and Nonalcoholic Steatohepatitis (NASH) also affected glycemic control and control of lipid profiles [45]. In addition, dietary intervention had a significant effect on patients’ FPG and lipid profiles [46]. Diet especially played an important role in the treatment of T2D. An excessive intake of saturated fatty acids (SFAs) would increase lipid profiles, while polyunsaturated fatty acids (PUFAs) have the opposite effect [47]. PUFA omega-3 supplements can reduce raised triglycerides [48]. Due to their rich dietary fiber and natural antioxidants [49], whole grains, fruits, vegetables, legumes, and nuts have been demonstrated to improve glycemic control and lipid profiles in patients with T2D [50]. Differences in the above characteristics of T2D may result in a different relationship between glycemic control and lipid profiles.

Different from previous studies that have used HbA1c as an indicator of glycemic control, in this study FPG was used as an indicator of glycemic control. Undoubtedly, HbA1c is the best indicator of glycemic control rather than FPG. However, HbA1c did not show glycemic variability over a period of time [15]. Changes in lipid profiles were immediately reflected in FPG. FPG is an economical and important indicator for the diagnosis and dynamic monitoring of T2D. During the control of lipid profiles and glycemic control in T2D patients, the relationship between changes in lipid profiles and FPG should be paid attention to. In order to manage T2D patients better, it is necessary to analyze the relationship between FPG and lipid profiles in time to guide the health management of T2D patients. Further experimental studies on the relationship between FPG and lipid profiles should be implemented in order to provide evidence for the management of T2D.

Our research had several limitations. First, due to the lack of some information, a large number of T2D patients were lost, which may be biased. Second, lack of information on diet, medication, and comorbidities in patients with T2D may also have an impact on the results. Third, the laboratory tests of T2D patients were not in the same hospital, which may also affect the results.

## 5. Conclusions

This study assessed the relationships between FPG control and lipid profiles in T2D patients in Qingdao, China. Our results showed that FPG was significantly associated with HDL and TC and was not associated with LDL and TG with adjustment for different confounding factors. Our findings suggested that TC and HDL should be focused on in the process of T2D health management.

## Figures and Tables

**Table 1 ijerph-17-05317-t001:** Summary statistics of characteristics according to gender (*M*(*Q*)).

Characteristic	Males (*n* = 2188)	Females (*n* = 3634)	Total (*n* = 5822)
Age (years)	67.00 (16.00)	67.00 (14.00)	67.00 (15.00)
BMI (kg/m^2^)	24.22 (3.53)	24.61 (3.94)	24.34 (3.96)
SBP (mmHg)	140.00 (18.00)	140.00 (20.00)	140.00 (20.00)
DBP (mmHg)	84.00 (10.00)	84.00 (11.00)	84.00 (10.00)
TG (mmol/L)	1.17 (0.82)	1.24 (0.90)	1.22 (0.86)
TC (mmol/dL)	5.07 (1.45)	5.20 (1.61)	5.13 (1.55)
HDL-C (mmol/dL)	1.30 (0.41)	1.30 (0.42)	1.30 (0.42)
LDL-C (mmol/dL)	2.56 (1.10)	2.59 (1.13)	2.56 (1.11)
FPG (mmol/L)	7.50 (1.97)	7.60 (2.00)	7.60 (2.00)
Duration of T2D (years)	1.93 (3.60)	2.16 (3.79)	2.06 (3.70)
Smoking, *n* (%)	737 (33.70)	176 (4.80)	913 (15.7)
Alcohol drinking, *n* (%)			
Not drinking	1425 (65.10)	3591 (98.80)	5016 (86.2)
Occasionally	387 (17.70)	25 (0.70)	412 (7.1)
Often	376 (17.20)	18 (0.50)	394 (6.7)
Hypertension, *n* (%)	1326 (60.60)	2258 (62.10)	3584 (61.6)
Antidiabetics			
No medication	1040 (47.50)	1384 (38.10)	2424 (41.6)
Taking medicine	707 (32.30)	1367 (37.60)	2074 (35.6)
Taking two types of medicine	423 (19.30)	835 (23.00)	1258 (21.6)
Taking three types of medicine	18 (0.80)	48 (1.30)	66 (1.1)
In marriage, *n* (%)	1860 (85.00)	2888 (79.50)	4748 (81.60)
Educational level			
Uneducated	385 (17.60)	1454 (40.00)	1839 (31.60)
Primary or junior high school	1531 (70.00)	2058 (56.60)	3589 (61.60)
High school and above	272 (12.40)	122 (3.40)	394 (6.80)
Physical exercise frequency			
Not exercising	1230 (56.20)	2226 (61.30)	3456 (59.40)
1–3 times per month	115 (5.30)	176 (4.80)	291 (5.00)
1–2 times per week	220 (10.10)	347 (9.50)	576 (9.70)
≥3 times per week	623 (28.50)	885 (24.40)	1508 (25.90)

*M*, median; *Q*, quartile range; BMI, body mass index; SBP, systolic blood pressure; DBP, diastolic blood pressure; FPG, fasting plasma glucose; TG, triglyceride; TC, total cholesterol; HDL-C, high density lipoprotein cholesterol; LDL-C, low density lipoprotein cholesterol.

**Table 2 ijerph-17-05317-t002:** Summary statistics and comparison of characteristics according to glycemic control (*M*(*Q*)).

Characteristic	FPG < 7.0 (*n* = 1602)	FPG ≥ 7.0 (*n* = 4220)	*p*
Age (years)	69.00 (13.00)	66.00 (15.00)	<0.001 **
BMI (kg/m^2^)	24.44 (3.91)	24.28 (3.94)	0.939
SBP (mmHg)	140.00 (16.00)	140.00 (20.00)	<0.001 **
DBP (mmHg)	80.50 (12.00)	85.00 (10.00)	<0.001 **
TG (mmol/L)	1.23 (0.87)	1.22 (0.86)	0.839
TC (mmol/L)	5.09 (1.51)	5.16 (1.50)	<0.001 **
HDL-C (mmol/L)	1.35 (0.40)	1.30 (0.40)	<0.001 **
LDL-C(mmol/L)	2.56 (1.10)	2.59 (1.11)	0.254
Duration of T2D(years)	1.91 (3.66)	2.15 (3.71)	0.059
Male, *n* (%)	609 (38.00)	1579 (37.40)	0.674
Smoking, *n* (%)	248 (15.50)	665 (15.80)	0.795
Alcohol drinking, *n* (%)			0.433
Not drinking	1386 (86.50)	3630 (86.00)	
Occasionally	118 (7.40)	294 (7.00)	
Often	98 (6.10)	296 (7.00)	
Hypertension, *n* (%)			<0.001 **
Antidiabetics			<0.001 **
No medication	843 (52.60)	1581 (37.50)	
Taking medicine	507 (31.60)	1567 (37.10)	
Taking two types of medicine	240 (15.00)	1018 (24.10)	
Taking three types of medicine	12 (0.70)	54 (1.30)	
In marriage, *n* (%)	1281 (80.00)	3467 (82.20)	0.054
Educational level			<0.001 **
Uneducated	561 (35.00)	1278 (30.30)	
Primary or junior high school	893 (55.70)	2696 (63.90)	
High school and above	148 (9.20)	246 (5.80)	
Physical exercise frequency			0.216
Not exercising	973 (60.70)	2483 (58.80)	
1–3 times per month	74 (4.60)	217 (5.10)	
1–2 times per week	166 (10.40)	401 (9.50)	
≥3 times per week	389 (24.30)	1119 (26.50)	

*M*, median; *Q*, quartile range; BMI, body mass index; SBP, systolic blood pressure; DBP, diastolic blood pressure; FPG, fasting plasma glucose; TG, triglyceride; TC, total cholesterol; HDL-C, high density lipoprotein cholesterol; LDL-C, low density lipoprotein cholesterol. **: *p* < 0.01.

**Table 3 ijerph-17-05317-t003:** Summary control and comparison of lipid profiles according to glycemic control (*n* (%)).

Lipid Profiles	FPG < 7.0 (1602 (27.50))	FPG ≥ 7.0 (4220 (72.48))	Total	*p*
TG (mmol/L)				0.532
<1.70	1180 (73.70)	3074 (72.80)	4254 (73.10)	
≥1.70	422 (26.30)	1146 (27.20)	1568 (26.90)	
TC (mmol/L)				<0.001 **
<4.50	560 (35.00)	1077 (25.50)	1637 (28.10)	
≥4.50	1042 (65.00)	3143 (74.50)	4185 (71.90)	
HDL-C (mmol/L)				<0.001 **
>1.00 or >1.30	1129 (70.50)	2610 (61.80)	3739 (64.20)	
≤1.00 or ≤1.30	473 (29.50)	1610 (38.20)	2083 (35.80)	
LDL-C (mmol/L)				0.117
<2.6 or <1.8	745 (46.50)	1866 (44.20)	2611 (44.80)	
≥2.6 or ≥1.8	857 (53.50)	2354 (55.80)	3211 (55.20)	

FPG, fasting plasma glucose; TG, triglyceride; TC, total cholesterol; HDL-C, high density lipoprotein cholesterol; LDL-C, low density lipoprotein cholesterol. **: *p* < 0.01.

**Table 4 ijerph-17-05317-t004:** Odd ratios (ORs) and their 95% confidence intervals (CI) of the risk of lipid profiles for good glycemic control.

Characteristic	Model 1	Model 2	Model 3	Model 4
TG (mmol/L)	0.984 (0.932, 1.039)	0.978 (0.926, 1.034)	0.966 (0.913, 1.022)	0.989 (0.935, 1.046)
TC (mmol/L)	0.853 (0.816, 0.891)	0.835 (0.798, 0.873)	0.840 (0.803, 0.879)	0.862 (0.823, 0.903)
LDL-C (mmol/L)	0.998 (0.932, 1.068)	0.981 (0.916, 1.051)	0.967 (0.902, 1.036)	0.987 (0.920, 1.060)
HDL-C (mmol/L)	2.229 (1.817, 2.735)	2.233 (1.818, 2.743)	2.223 (1.809, 2.733)	2.173 (1.761, 2.683)
Age (years)		1.022 (1.016, 1.027)	1.021 (1.015, 1.028)	1.023 (1.017, 1.030)
Educational level				
Uneducated			reference	reference
Primary or junior high school			0.892 (0.775, 1.026)	0.898 (0.778, 1.037)
High school and above			1.603 (1.257, 2.044)	1.511 (1.174, 1.945)
Physical exercise frequency				
Not exercising			reference	reference
1–3 times per month			0.926 (0.697, 1.230)	0.801 (0.600, 1.070)
1–2 times per week			1.018 (0.834, 1.242)	1.038 (0.847, 1.272)
≥3 times per week			0.853 (0.741, 0.982)	0.802 (0.694, 0.926)
Hypertension, *n* (%)				
no				reference
yes				0.559 (0.493, 0.634)
Antidiabetics				
No medication				reference
Taking medicine				0.584 (0.510, 0.669)
Taking two types of medicine				0.438 (0.370, 0.519)
Taking three types of medicine				0.487 (0.256, 0.925)

TG, triglyceride; TC, total cholesterol; HDL-C, high density lipoprotein cholesterol; LDL-C, low density lipoprotein cholesterol. Model 1: unadjusted. Model 2: adjusted by age and gender. Model 3: adjusted by age, gender, smoking, alcohol drinking, marital status, educational level, and physical exercise frequency. Model 4: adjusted by age, gender, smoking, alcohol drinking, marital status, educational level, physical exercise frequency, BMI, hypertension, duration of T2D, and antidiabetics.

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
