# Peer review of "Relationship between Lipid Profiles and Glycemic Control Among Patients with Type 2 Diabetes in Qingdao, China"

_ijerph, 2020, doi:10.3390/ijerph17155317_

Round 1
Reviewer 1 Report
The manuscript reviewed focuses upon the relationship between lipid profiles and glycemic control in people with T2D.
The manuscript is well-written and easy to follow but I do have some questions and concern
- My main question is: what is the novelty of the study? Not sure there is a lot of new information provided
- Other major concern is that fasting plasma glucose can by no means be considered an indicator of glycemic control.
Other comments:
- Intro, line 33 – why physical and mental pain – not sure the word pain is the most appropriate
- Lines 46-47 : not sure this study is the best to support this statement
- There are reviews on the topic dyslipidemia and T2D and some of these should be included
- Further the importance of good glycemic control and controls of lipids has been assessed in several studies and these should be referred to and why not some of the CVOT in relation to CVD in T2D subjects. I do not think the literature referred to is the most representative.
- A large number of the original sample size is lost 15369 vs 5822
- Table 1 gives baseline characteristics in the total sample and men and women separately – no p-values are needed
- Table 2 SBP difference between the groups – p-value <0.001 – is this really the case ?
And even if this is the case this is clinically meaningless – since there is no difference
- Also the relevance of the findings in the other tables must be considered from a clinical point of view – a small p-value is not the major thing.
Author Response
Dear Reviewer,
Thanks for your valuable comments and suggestion. These comments and suggestions from you have led to great improvement of our manuscript. We have revised the original manuscript according to all these insightful comments. Point-by-point responses are attached and given below. We hope we have addressed all the concerns satisfactorily.
Thanks again for your great help, and we are looking forward to your reply.
Best regards.
Shukang Wang
Response to Reviewer 1
The manuscript is well-written and easy to follow but I do have some questions and concern
- My main question is: what is the novelty of the study? Not sure there is a lot of new information provided
Response: Thank you very much for pointing this out, we should indeed explain this more clearly. Many studies showed that glycemic control and the control of lipid profiles are very important and there is an interaction between the two in T2DM patients. Previous many studies in which HbA1c was used as an indicator of glycemic control have shown controversial results for the relationship between glycemic control and lipid profiles in patients with T2DM. However, HbA1c is a stable indicator of glycemic control that reflects an average glycemic level of about three months and does not show glycemic variability over a period of time. Fasting plasma glucose (FPG) is an economical and important indicator for the diagnosis and glycemic variability of T2D. Few studies have shown the relationship between FPG control and lipid profiles. Therefore, this study aims to assess the levels of lipid profiles and FPG, and investigate the relationship between FPG control and lipid profiles in patients with T2D in Qingdao City, Shandong Province, China. In addition, it is hoped that more studies on the relationship between FPG control and lipid profiles will be compared with this study in the future. In this study, the points we want to express are as follows: HbA1c is the best indicator of glycemic control rather than FPG. Changes in lipid profiles are immediately reflected in FPG. During the control of lipid profiles and glycemic control in type 2 diabetes patients, the relationship between changes in lipid profiles and FPG should be paid attention to. In order to manage type 2 diabetes patients better, it is necessary to analyze the relationship between FPG and lipid profiles in time to guide practice. The corresponding content has been added to the preface discussion and conclusion (see line 60-67 and line 237-286 of the revised manuscript).
- Other major concern is that fasting plasma glucose can by no means be considered an indicator of glycemic control.
Response: Thanks for pointing this out. We really should discuss this in more detail and express our views more clearly. HbA1c rather than FPG should be used as an indicator of glycemic control. However, HbA1c does not show glycemic variability during a period of time. FPG is an economical and important indicator for the diagnosis and glycemic variability of T2D. And changes in lipid profiles are immediately reflected in FPG. In fact, in this study we wants to express that more attention should be paid to the relationship between FPG control and lipid profiles during the treatment of Type 2 diabetes patients. We have made corresponding changes in the discussion and conclusion section (see line 237-286 of the revised manuscript).
Other comments:
- Intro, line 33 – why physical and mental pain – not sure the word pain is the most appropriate
Response: Thank you for pointing this out. At the same time, the expression here is too cumbersome, we have made corresponding changes (see line 35 of the revised manuscript).
- Lines 46-47: not sure this study is the best to support this statement
Response: Thank you for pointing this out. This study alone does not best support this statement. Many references have been cited here (see line 50 of the revised manuscript).
- There are reviews on the topic dyslipidemia and T2D and some of these should be included
Response: Thank you very much for your suggestion, we really should have a more thorough discussion. Some review studies on the topic dyslipidemia and T2D are also included in this study (see line 244-259 of the revised manuscript).
- Further the importance of good glycemic control and controls of lipids has been assessed in several studies and these should be referred to and why not some of the CVOT in relation to CVD in T2D subjects. I do not think the literature referred to is the most representative.
Response: Thank you very much for your suggestion. Some more representative literatures have been cited (see line 250-259 of the revised manuscript).
- A large number of the original sample size is lost 15369 vs 5822
Response: Thank you for pointing this out. Unfortunately, due to the absence of information a large number of the original sample size is lost, which is also added to the limitations of this study (see line 269-270 of the revised manuscript).
- Table 1 gives baseline characteristics in the total sample and men and women separately – no p-values are needed
Response: Thank you for pointing this out. According to your suggestion, we have made the corresponding changes (see Table 1 and line 138-141 of the revised manuscript).
- Table 2 SBP difference between the groups – p-value <0.001 – is this really the case?
And even if this is the case this is clinically meaningless – since there is no difference
Response: Thank you for pointing this out. We have rechecked it and the SBP result in Table 2 is correct. The median SBP was 140 in the two groups (FPG≥7.0 and FPG<7.0) compared, while the average SBP was 141.72 and 138.99, respectively. Because the data is skewed, the median is used to describe the average of the data in this study.
- Also the relevance of the findings in the other tables must be considered from a clinical point of view – a small p-value is not the major thing.
Response: Thank you very much for your comment. The difference is statistically significant and not necessarily clinically significant. We must also consider this from a clinical point of view.

Reviewer 2 Report
The manuscript is interesting and provides important information regarding metabolic parameters (glycemia and plasma lipids) in patients with diabetes mellitus. Furthermore, these investigations allow the initiation of other studies to improve metabolic control in patients with diabetes mellitus. However I have the following comments.
I. Major Comments:
1. In the discussion, I suggest including a paragraph regarding the dietary aspects that are directly related to an increase in glycemia and plasma lipids (TG, TC and LDL-C). Specifically, refined carbohydrates (sugar) and saturated fatty acids (palmitic acid). Along with indicating that the low intake of n-3 PUFA, dietary fiber and natural antioxidants is related to an increase in glycemia and plasma lipids (TG, TC and LDL-C).
Suggested reference:
Relevant Aspects of Nutritional and Dietary Interventions in Non-Alcoholic Fatty Liver Disease. Int J Mol Sci. 2015 Oct 23; 16 (10): 25168-98. doi: 10.3390 / ijms161025168.
2. In the title and all the text I suggest using "type 2 diabetes" or "diabetes mellitus". both are the same
3. In the methodology section, information is lacking regarding the inclusion and exclusion criteria
4. In the study group, are there patients using insulin ?
5. Indicate the number of subjects "men" and "women"
II. Minor Comments:
1. Improve the writing of the study objective
2. Some sentences are very long, I suggest editing the text (improve understanding of the text)
Author Response
Dear Reviewer,
Thanks for your valuable comments and suggestion. These comments and suggestions from you have led to great improvement of our manuscript. We have revised the original manuscript according to all these insightful comments. Point-by-point responses are attached and given below. We hope we have addressed all the concerns satisfactorily.
Thanks again for your great help, and we are looking forward to your reply.
Best regards.
Shukang Wang
Response to Reviewer 2
Comments and Suggestions for Authors
The manuscript is interesting and provides important information regarding metabolic parameters (glycemia and plasma lipids) in patients with diabetes mellitus. Furthermore, these investigations allow the initiation of other studies to improve metabolic control in patients with diabetes mellitus. However I have the following comments.
- Major Comments:
- In the discussion, I suggest including a paragraph regarding the dietary aspects that are directly related to an increase in glycemia and plasma lipids (TG, TC and LDL-C). Specifically, refined carbohydrates (sugar) and saturated fatty acids (palmitic acid). Along with indicating that the low intake of n-3 PUFA, dietary fiber and natural antioxidants is related to an increase in glycemia and plasma lipids (TG, TC and LDL-C).
Suggested reference:
Relevant Aspects of Nutritional and Dietary Interventions in Non-Alcoholic Fatty Liver Disease. Int J Mol Sci. 2015 Oct 23; 16 (10): 25168-98. doi: 10.3390 / ijms161025168. Response: Thank you very much for your comments and suggestions. The influence of dietary factors should be discussed in detail. According to your suggestion, we have added corresponding content in the discussion section (see line 248-256 and line 265-266 of the revised manuscript).
- In the title and all the text I suggest using "type 2 diabetes" or "diabetes mellitus". both are the same
Response: Thanks for your suggestions. We have made corresponding changes in the title and all the text I.
- In the methodology section, information is lacking regarding the inclusion and exclusion criteria
Response: Thanks for pointing this out. The inclusion and exclusion criteria have been supplemented in the methodology section (see line 71-75 of the revised manuscript).
- In the study group, are there patients using insulin?
Response: Thanks for mentioning this. There are seven patients using insulin in this study. When we eliminated these seven patients and analyzed the data again, the results did not change except for the third place after the decimal point of some OR value.
- Indicate the number of subjects "men" and "women"
Response: Thanks for mentioning this again. The number of subjects "men" and "women" were 2188 and 3634 respectively. Due to the good compliance of female patients, there are more women with complete information than men in the health management system. This is also one of the limitations of this study (see line 269-270 of the revised manuscript).
- Minor Comments:
- Improve the writing of the study objective
Response: Thank you very much for your comment. We have made corresponding changes (see line 13-15 of the revised manuscript).
- Some sentences are very long, I suggest editing the text (improve understanding of the text)
Response: Thank you very much for your comments and suggests. We have made corresponding changes in the full text.

Reviewer 3 Report
Method section. Started off well but not enough info on methodology e.g. make of BP machine, how lipids were measured, on what platform and according to what standards. Also tube type, storage prior to analysis etc.
you start of with HBA1C and finish with HbA1C.
i don’t know why you have compared men and women. It does show remarkably quite a stark difference between the sexes and how much iller the women are.
i am not sure why you keep mentioning the hdl infusion study. Also not sure I agree with your conclusion, how have you proven that fpg is a marker of choice?
I think it is a bit simplistic I your discussion overall. Rest is all nicely written. We know that not all those with type 2 DM have the same disease e.g. some with fasting hyperglycaemia and those with post principal hyperglycaemia. High trigs, low HDL and DM are all part of the metabolic syndrome which is well described and I don’t think you have disproven that link. I suspect you have a mixed group with different metabolic status and may be this is best as a demonstration of severity and control of DM in this population and that drugs that increase HDL may be mor pertinent than those that lower trigs. Of course we have shown that high HDL doesn’t matter it is only low that is bad. I wonder if selecting only those with low HDL, ie the met syndrome cohort, whether you would see that high trigs correlate with FPG as expected. Trigs you would expect to change quickly with glucose but HDL to be more fixed like HbA1c or would you?
nicely written and clear that you have done a lot of work I just don’t think the focus is quite right and your conclusions therefore I do not agree with.
Author Response
Dear Reviewer,
Thanks for your valuable comments and suggestion. These comments and suggestions from you have led to great improvement of our manuscript. We have revised the original manuscript according to all these insightful comments. Point-by-point responses are attached and given below. We hope we have addressed all the concerns satisfactorily.
Thanks again for your great help, and we are looking forward to your reply.
Best regards.
Shukang Wang
Response to Reviewer 3
Comments and Suggestions for Authors
- Method section. Started off well but not enough info on methodology e.g. make of BP machine, how lipids were measured, on what platform and according to what standards. Also tube type, storage prior to analysis etc.
Response: Thank you very much for pointing this out, we really should express these in more detail. We have added these in the method section (see line 96-99 of the revised manuscript).
- you start of with HBA1C and finish with HbA1C.
Response: Thank you very much for pointing this out. We changed HBA1C to HbA1C in full text.
- i don’t know why you have compared men and women. It does show remarkably quite a stark difference between the sexes and how much iller the women are.
Response: Thank you very much for pointing this out. Gender is included in the final multivariate analysis. There is really no need to make a comparison between the sexes, so the result of the comparison has been deleted (see table 1 of the revised manuscript).
- i am not sure why you keep mentioning the hdl infusion study. Also not sure I agree with your conclusion, how have you proven that fpg is a marker of choice?
Response: Thank you very much for pointing this out. In the discussion, I did not express our points clearly. The HDL infusion study is cited again to illustrate that changes in HDL immediately cause changes in fasting plasma glucose (FPG). HbA1c is the best indicator of glycemic control rather than FPG. In this study, the points I want to express are as follows: FPG is an economical and important indicator for the diagnosis and dynamic monitoring of T2D. Changes in lipid profiles are immediately reflected in FPG. During the control of lipid profiles and glycemic control in type 2 diabetes patients, the relationship between changes in lipid profiles and FPG should be paid attention to. In order to manage type 2 diabetes patients better, it is necessary to analyze the relationship between FPG and lipid profiles in time to guide practice. The corresponding content has been added to the preface discussion and conclusion (see line 237-284 of the revised manuscript).
- I think it is a bit simplistic I your discussion overall. Rest is all nicely written. We know that not all those with type 2 DM have the same disease e.g. some with fasting hyperglycaemia and those with post principal hyperglycaemia. High trigs, low HDL and DM are all part of the metabolic syndrome which is well described and I don’t think you have disproven that link. I suspect you have a mixed group with different metabolic status and may be this is best as a demonstration of severity and control of DM in this population and that drugs that increase HDL may be mor pertinent than those that lower trigs. Of course we have shown that high HDL doesn’t matter it is only low that is bad. I wonder if selecting only those with low HDL, ie the met syndrome cohort, whether you would see that high trigs correlate with FPG as expected. Trigs you would expect to change quickly with glucose but HDL to be more fixed like HbA1c or would you?
Response: Thank you very much for your comments, you are right. Our discussion is indeed too simple. We do not know the other diseases that these diabetic patients suffer from at the same time, the detailed drug use of the patient and the diet of the patient. These factors will affect the results. This research population is indeed a mixed population with different metabolic states, and there are many problems that need to be further explored in future research. We will add this part in the discussion (see line 243-268 of the revised manuscript).
- nicely written and clear that you have done a lot of work I just don’t think the focus is quite right and your conclusions therefore I do not agree with.
Response: Thank you very much for your comments again. We did overlook many problems when we came to the conclusion. In this study, the points we want to express are as follows: HbA1c is the best indicator of glycemic control rather than FPG. FPG is an economical and important indicator for the diagnosis and dynamic monitoring of T2D. Changes in lipid profiles are immediately reflected in FPG. During the control of lipid profiles and glycemic control in type 2 diabetes patients, the relationship between changes in lipid profiles and FPG should be paid attention to. In order to manage type 2 diabetes patients better, it is necessary to analyze the relationship between FPG and lipid profiles in time to guide practice. We have made corresponding changes in the discussion and conclusion section (see line 237-286 of the revised manuscript).

Round 2
Reviewer 1 Report
No further comments
Author Response
Dear Reviewer,
Thanks for your valuable comments and suggestion again. We have revised the original manuscript according to all these insightful comments. And we have invited English professionals to carefully check the style and spelling of the language again. We hope we have addressed all the concerns satisfactorily.
Thanks again for your great help, and we are looking forward to your reply.
Best regards.
Shukang Wang

Reviewer 3 Report
Thank you for all the changes. It makes more sense to me now and I understand better what you are trying to show.
Thank you for adding method details i would still say that is not enough as you are primarily looking at lipids having split by FPG - all laboratory results. I don't know if they were all measured in the same place and by what method(s) or analyser e.g. cholesterol was measured by cholesterol oxidase method on Roche Cobas, in a clinical laboratory accredited to ISo15189, How was LDL calculated etc...
Minor thing is now the word 'diabetics' has crept in and the patient associations don't like it, they say it is derogatory, so it should be 'people with diabetes' or something similar.
Thank you.
Author Response
Dear Reviewer,
Thanks for your valuable comments and suggestion again. We have revised the original manuscript according to all these insightful comments. And we have invited English professionals to carefully check the style and spelling of the language again. Point-by-point responses are attached and given below. We hope we have addressed all the concerns satisfactorily.
Thanks again for your great help, and we are looking forward to your reply.
Best regards.
Shukang Wang
Response to Reviewer 3
Comments and Suggestions for Authors
Thank you for all the changes. It makes more sense to me now and I understand better what you are trying to show.
- Thank you for adding method details i would still say that is not enough as you are primarily looking at lipids having split by FPG - all laboratory results. I don't know if they were all measured in the same place and by what method(s) or analyser e.g. cholesterol was measured by cholesterol oxidase method on Roche Cobas, in a clinical laboratory accredited to ISo15189, How was LDL calculated etc...
Response: Thank you very much for pointing this out. We should indeed express this more clearly. All tests are done in different hospitals with clinical laboratories accredited to ISo15189. This may have an impact on the results. This has also been added to the limitations (see line 277-278 of the revised manuscript). Generally, the serum samples were tested within two hours. If the sample is not tested within two hours, it will be frozen at -20°C until testing. Despite different laboratories, the testing methods are the same. The detection methods have been supplemented in the method section (see line 97-103 of the revised manuscript).
- Minor thing is now the word 'diabetics' has crept in and the patient associations don't like it, they say it is derogatory, so it should be 'people with diabetes' or something similar.
Response: Thank you very much for pointing this out. We have changed the word ‘diabetics’ to ‘T2D patients’ in the full text (see line 272 of the revised manuscript).
